# Ionization of Decamethylmanganocene: Insights from the DFT-Assisted Laser Spectroscopy

**DOI:** 10.3390/molecules27196226

**Published:** 2022-09-22

**Authors:** Sergey Ketkov, Sheng-Yuan Tzeng, Elena Rychagova, Wen-Bih Tzeng

**Affiliations:** 1G.A. Razuvaev Institute of Organometallic Chemistry of the Russian Academy of Sciences, 49 Tropinin St., 603950 Nizhny Novgorod, Russia; 2Institute of Atomic and Molecular Sciences, Academia Sinica, 1 Section 4, Roosevelt Road, Taipei 10617, Taiwan

**Keywords:** metallocenes, ionization energy, laser spectroscopy, DFT calculations, electron density

## Abstract

Metallocenes represent one of the most important classes of organometallics with wide prospects for practical use in various fields of chemistry, materials science, molecular electronics, and biomedicine. Many applications of these metal complexes are based on their ability to form molecular ions. We report the first results concerning the changes in the molecular and electronic structure of decamethylmanganocene, Cp*_2_Mn, upon ionization provided by the high-resolution mass-analyzed threshold ionization (MATI) spectroscopy supported by DFT calculations. The precise ionization energy of Cp*_2_Mn is determined as 5.349 ± 0.001 eV. The DFT modeling of the MATI spectrum shows that the main structural deformations accompanying the detachment of an electron consist in the elongation of the Mn-C bonds and a change in the Me out-of-plane bending angles. Surprisingly, the DFT calculations predict that most of the reduction in electron density (ED) upon ionization is associated with the hydrogen atoms of the substituents, despite the metal character of the ionized orbital. However, the ED difference isosurfaces reveal a complex mechanism of the charge redistribution involving also the carbon atoms of the molecule.

## 1. Introduction

In 2022, chemists celebrate the 70th anniversary of the first publications on a unique sandwich structure of ferrocene, Cp_2_Fe (Cp = η^5^-C_5_H_5_) [1,2]. With the discovery of this iconic metal complex, a new era of organometallic chemistry has started. Currently, metallocenes are of ever-increasing interest among researchers in fundamental and applied science, since they are relevant to the modern theories of metal–ligand bonding [3,4,5,6], electro- and magnetochemistry [7,8,9,10,11,12,13], metallopolymers [14,15,16,17,18], biomedicine [19,20,21,22], catalysis [23,24,25], nanoelectronics [26,27,28,29,30,31,32] and molecular machines [33,34,35]. Many applications of metallocenes take advantage of their pronounced reducing properties, which allow for the easy formation of stable or reactive sandwich ions [7,8,9,10,14,15,16,23,24,25,26,27,28,29,30,33,34,35,36,37,38,39]. These properties are associated with the low ionization energies, *I*, corresponding to the detachment of an electron from a sandwich high occupied molecular orbital (MO) with substantial metal *d* character [40,41,42,43,44,45,46,47]. Comprehensive studies of ionization processes in metallocenes are, therefore, especially important for understanding their reactivity.

Unprecedented opportunities for studying the details of the ionization of molecules appeared with the development of high-resolution methods using laser irradiation. Precise measurements of ionization energies are provided by the zero kinetic energy (ZEKE) or mass-analyzed threshold ionization (MATI) spectroscopy [48,49] using the laser excitation of molecules to high Rydberg levels (so-called ZEKE states) which are then ionized by an electric pulse. The ZEKE states of polyatomic molecules with high *I* values are usually populated by multiphoton laser excitation through intermediate vibronic levels. For organometallics, this approach encounters experimental difficulties associated with poor resolution of vibronic states and fast photodissociation of the excited molecules [50,51]. However, the ZEKE states of metallocenes and related sandwich complexes with *I* < 6 eV can be populated in a one-photon experiment using standard laser sources.

Nevertheless, the experimental measurement of the MATI and ZEKE spectra of sandwich compounds remains a nontrivial task due to the low volatility and thermal stability of these metal complexes as well as their air and moisture sensitivity. Among metallocenes, the MATI spectrum has so far been recorded only for Cp_2_Co [52]. The MATI/ZEKE data have been obtained also for some bisarene complexes [53,54,55,56,57,58,59,60,61,62,63,64,65] and few mixed sandwiches [66]. These studies demonstrated new possibilities provided by the MATI and ZEKE techniques for detecting substituent effects and structural changes of sandwich molecules upon ionization [67]. In the present work, we report the first MATI spectrum of jet-cooled decamethylmanganocene, Cp*_2_Mn (Cp* = η^5^-C_5_Me_5_). The manganocene-methylated derivatives are of particular fundamental interest, since the introduction of the CH_3_ groups to the Cp rings of Cp_2_Mn leads to a “switching” of the ground electronic state from the ^6^*A*_1_ sextet to the ^2^*E*_2_ doublet [40,68,69,70,71,72,73] (we use the irreducible representations of the *D*_5_ point group for designation of the electronic states and MOs). In contrast to many other metallocenes, manganocene does not form stable cationic derivatives. On the other hand, its permethylated low-spin analogue, Cp*_2_Mn, is readily oxidized to yield the Cp*_2_Mn^+^ cation [11,73]. Such systems can display properties of bulk molecular ferromagnets [74,75]. Therefore, the details of structural and energetic changes accompanying the ionization of Cp*_2_Mn attract special attention. Our combined MATI/DFT study provides new insights into these details. 

## 2. Results and Discussion

Similar to other theoretical studies of metallocenes [72,76,77,78,79], our B3PW91/6-311++G(d,p) DFT calculations show that high-energy occupied MOs of Cp*_2_Mn have a predominantly metal *d* character (Figure 1). The electronic configurations of the Cp*_2_Mn^0^ neutral and the Cp*_2_Mn^+^ cation are …(*d*_x^2^-y^2^_)^1^(*d*_z^2^_)^2^(d_xy_)^2^ and …(*d*_x^2^-y^2^_)^1^(*d*_z^2^_)^1^(d_xy_)^2^, respectively. These configurations correspond to the ^2^*E*_2_ doublet ground state of the *D*_5_ neutral and the ^3^*E*_2_ triplet ground state of the ion [40,71]. However, the degeneracy of MO *d*_x^2^-y^2^_ and d_xy_, which would take place in the *D*_5_ point group, is actually lifted in Cp*_2_Mn (Figure 1) due to the Jahn–Teller (JT) distortion.

The ground-state ^3^*E*_2_ Cp*_2_Mn^+^ cation is formed by a detachment of the non-bonding *d*_z^2^_ β-electron from the neutral molecule. The *I* value can be determined in a laser ionization experiment by recording the photoionization efficiency (PIE) curve or, much more accurately, by measuring the MATI spectrum.

### 2.1. PIE and MATI Spectra of Cp*_2_Mn

The PIE curve of Cp*_2_Mn (Figure 2a) shows a rather sharp step in the 43,000–43,200 cm^−1^ region corresponding to the ionization energy of 5.33–5.36 eV, which agrees very well with the *I* = 5.33 ± 0.10 eV value determined earlier from the “classical” photoelectron spectrum [71]. The MATI spectrum of Cp*_2_Mn (Figure 2b) reveals a clearly defined peak arising from a detachment of an electron from the ZEKE levels lying a few wavenumbers below the cation ground vibronic state. The *I* value of 43,146 ± 8 cm^−1^, or 5.349 ± 0.001 eV, corresponds to a narrow MATI signal descend range. Therefore, the MATI spectrum of Cp*_2_Mn provides an extremely accurate ionization energy, the error being two orders of magnitude smaller than that of standard photoelectron spectroscopy. 

The precise *I* value of Cp*_2_Mn obtained from the MATI spectrum provides new accurate information on the difference between the Mn-Cp* bond dissociation energies in the decamethylmanganocene cation and neutral [67]:*D*(Cp*_2_Mn ^+^) − *D*(Cp*_2_Mn ^0^) = *I*(Mn^0^) − *I*(Cp*_2_Mn^0^),(1)
where *D*(Cp*_2_Mn ^+/0^) corresponds to the energy change in the reaction:Cp*_2_Mn^+/0^ → Mn^+/0^ + 2C_5_Me_5_^●^.(2)

Taking into account the known ionization energy of the manganese atom, *I*(Mn^0^) = 4.43402 eV [80], and the *I*(Cp*_2_Mn^0^) value derived from our MATI spectrum we obtain the dissociation energy difference (Equation (1)) as 2.085 ± 0.001 eV, or 201.2 ± 0.1 kJ mol^−1^. Therefore, the mean Mn-Cp* bond dissociation energy in the decamethylmanganocene cation, equal to 0.5 *D*(Cp*_2_Mn ^+^), is 100.6 kJ mol^−1^ larger than that in the neutral molecule. The accuracy in the determination of this difference from the MATI data presented in this work (0.1 kJ mol^−1^) is much better than that obtained from the UV photoelectron spectrum of Cp*_2_Mn [71] (ca. 10 kJ mol^−1^). As a result, the use of laser threshold ionization spectroscopy provides new possibilities for studying fine substituent effects in manganocenes, similar to those found for bisarene sandwich systems [67]. 

The MATI spectrum of Cp*_2_Mn (Figure 2b) reveals no lower-energy vibronic features accompanying the strong peak with the maximum at 43,100 cm^−1^. Therefore, this peak corresponds to the origin (the 0_0^0^_ transition between the ground vibrational states of Cp*_2_Mn^0^ and Cp*_2_Mn ^+^). Then, the vertical and adiabatic ionization energies of Cp*_2_Mn coincide. A different situation holds with the MATI spectrum of Cp_2_Co [51], where the vertical *I* value is 0.115 eV higher than the adiabatic ionization energy. This difference can be associated with the antibonding nature of the ionized MO in Cp_2_Co [40,51,77]. On the other hand, the chromium bisarene complexes with the non-bonding *d*_z^2^_ electron ionized produce MATI spectra [52,53,54,55,56,58,64,65,66] showing the 0_0^0^_ vibronic component as the strongest feature like Cp*_2_Mn. The vibrationally resolved high-resolution MATI spectra provide much more information on the sandwich molecules than structureless “classical” photoelectron spectra or PIE curves.

### 2.2. Vibronic Structure of the Cp*_2_Mn MATI Spectrum

Detailed analysis of the Cp*_2_Mn MATI spectrum recorded at a higher concentration of the organometallic molecules in the supersonic jet reveals the weaker high-energy vibronic components accompanying the origin (Figure 3a). Their assignment can be made on the basis of the DFT modeling. The B3PW91/6-311++G(d,p) simulation (Figure 3b) gives a MATI vibronic structure that agrees very well with the experiment. The relative positions of the vibronic components (Table 1) correspond to the vibrational frequencies of the free Cp*_2_Mn^+^ cation. Only two normal vibrations of Cp*_2_Mn^+^ are responsible for the MATI features of noticeable intensity (Figure 3). They are denoted in this work as ν_1_ and ν_2_. According to the DFT calculations, the 130 cm^−1^ ν_1_ mode corresponds to the Me out-of-plane bend, while the 352 cm^−1^ ν_2_ vibration represents the symmetric Mn-ring stretch without displacement of the methyl carbon nuclei. Both the ν_1_ and ν_2_ modes involve the change of the Mn-C_ring_-C_Me_ angle. The relative atomic shifts for the ν_1_ and ν_2_ vibrations are shown in Figure 4, and the corresponding animations are presented in the Appendix A. 

Both ν_1_ and ν_2_ would be totally symmetric modes in a *D*_5_ decamethylmetallocene molecule like closed-shell Cp*_2_Fe. The experimental and simulated MATI spectra of Cp*_2_Mn (Figure 3) do not contain vibronic components that could be associated with the JT distortion of the neutral or cationic species in the degenerate *E*_2_ electronic state. This testifies for similar shifts of the Cp*_2_Mn^0^ and Cp*_2_Mn^+^ molecular geometries from the *D*_5_ point group along the JT active vibrational coordinates. The good agreement between the experimental and simulated MATI vibronic structure (Figure 3, Table 1) shows that the B3PW91/6-311++G(d,p) level of DFT provides reliable information about the structural changes in Cp*_2_Mn upon ionization. 

### 2.3. Changes in the Geometry of Decamethylmanganocene upon Ionization

The structural transformations accompanying ionization of Cp*_2_Mn can be analyzed on the basis of the neutral and ion geometries obtained from the X-ray diffraction experiments [68,74]. However, in this case the changes caused by the intramolecular effects would be superimposed on those resulted from the intermolecular and cation–anion interactions. The Cp*_2_Mn structure in the gas phase has been determined by electron diffraction [69], but these data provide only averaged geometry of the sandwich molecule corresponding to the high symmetry of the *D*_5d_ point group. Therefore, the DFT calculations verified by the simulation of the experimental gas-phase MATI spectrum would be preferable as a basis for studying the fine details of the structural changes accompanying ionization of the Cp*_2_Mn molecule. 

The atom numbering as well as selected bond lengths and angles in one Cp*Mn fragment are given in Figure 5 and Figure 6. The atomic coordinates are provided in the Appendix A). The geometry of the other fragment is identical since the molecule with a staggered conformation of the Cp* ligands possesses a center of inversion. The optimized staggered geometry of Cp*_2_Mn^0^ is in accord with the X-ray structure [68,81] (Appendix A, see the Appendix A). The calculated interatomic distances in Cp*_2_Mn^0^ (Figure 5a) agree well with the single-crystal X-ray diffraction data [68] (*r*(C-C) = 1.409–1.434 Å for the carbocycle and *r*(Mn-C) = 2.105–2.118 Å) and gas-phase electron diffraction results [69] (1.440 and 2.130 Å, respectively). The DFT-based structural parameters of the Cp*_2_Mn^+^ ion (Figure 5b) also correlate with the crystal structure [74], though the experimental solid-phase parameters depend strongly on the counter ion.

The optimized structure of the neutral Cp*_2_Mn molecule or the corresponding cation is nearly symmetric relative to the MnC1C6 plane. Together with the inversion center at the Mn atom, this means that the molecular symmetry is close to the *C*_2h_ point group. The deviation from the ideal *D*_5d_ symmetry is caused by the JT distortion of the Cp*_2_Mn neutral or ion in the degenerate *E*_2_ ground electronic state. The JT distortion in both Cp*_2_Mn^0^ and Cp*_2_Mn^+^ includes mainly: (1) a slippage of the Cp* ligands leading to a shift of the Mn atom projection on the ring plane from the ring centroid and to different Mn-C bond lengths; (2) an in-plane ring deformation (a contraction of the C1-C2 and C1-C5 bonds, an elongation of the C3-C4 bond); (3) unequal bending of the methyl groups away from the metal atom (the C8 and C9 atoms lie at the longest distance from the ring plane). The first and the second trend were also revealed by the X-ray study [68]. 

It should be noticed that the JT distortion of Cp*_2_Mn^0^ differs from that in Cp_2_Co^0^ where the Cp rings become non-planar [51,82]. This difference arises from the different nature of the single-occupied MO in these sandwich molecules. In Cp*_2_Mn^0^, the unpaired electron occupies MO *d*_x^2^-y^2^_ belonging to the *e*_2_ set in a non-distorted molecule. In Cp_2_Co^0^, the single-occupied MO is derived from the Co 3d_xz_ or 3d_yz_ and Cp π(*e*_1_) wavefunctions.

The structural transformation of Cp*_2_Mn^0^ upon ionization consists mainly in lengthening the Mn-Cp* distances and changing the Me bending angles (Table 2). As a result of the electron detachment, the distance between the ligand ring planes increases by 0.048 Å, from 3.494 Å to 3.542 Å. This increase in the ligand–ligand distance exceeds that computed earlier for (η^6^-C_6_H_6_)_2_Cr (0.038 Å [66]). The symmetry of the Cp*_2_Mn^0^ neutral and Cp*_2_Mn^+^ ion is the same, so only totally symmetric vibrations ν_1_ and ν_2_ corresponding to the Me bend and Mn-Cp* stretch, respectively, are excited in the MATI experiment. In contrast to decamethylmanganocene, ionization of cobaltocene leads to the highly symmetric closed-shell Cp_2_Co^+^ ion. Accordingly, the MATI spectrum of Cp_2_Co reveals the *e*_2_ and *e*_1_ vibrational modes which are active in the JT effect and pseudo-JT effect occurring in the neutral molecule [51,83]. The ionized MO of Cp*_2_Mn^0^ represents an almost “pure” 3*d*_z^2^_ orbital of the Mn atom (Figure 1). However, the electron density (ED) analysis reveals a much more complex charge redistribution upon ionization of decamethylmanganocene than would be expected from the simple subtraction of the *d*_z^2^_ ED.

### 2.4. Charge Density Redistribution after Electron Detachment from Cp*_2_Mn^0^


The ED difference (EDD) isosurfaces (Figure 7) corresponding to the ED change on going from Cp*_2_Mn^0^ to Cp*_2_Mn^+^ at the geometry of the neutral (the vertical ionization) reveal areas of both a decrease and an increase in the ED. The ED gain near the metal is the result of charge transfer from the Cp* ligands to the Mn atom during ED relaxation after ionization. The 0.005 a.u. EDD isosurface (Figure 7a) shows the ED loss arising from the detachment of the Mn *d*_z^2^_ electron and from the consequent transfer of the ligand π-electron density to the Mn atom. The largest area of the positive EDD visualizes the acquisition of ED by the metal atom through the Mn(d_xz,yz_)/Cp(π) *e*_1_-occupied orbital. A smaller ED shift from C(Me) to C(ring) along the C-C σ-bonds is also observed (Figure 7a). The 0.003 a.u. and 0.001 EDD isosurfaces (Figure 7b) reveal an additional ED loss by the H atoms of the Me groups accompanied by a ED increase near C(Me) around the C-H bonds. The positive EDD region appears inside the carbocycles near the C-C σ-bonds.

The inhomogeneous ED redistribution upon ionization of decamethylmanganocene leads to nontrivial changes in the atomic charges. They were calculated in this work by integration of the ED over the atomic basins within the frames of the Quantum Theory of Atoms in Molecules (QTAIM) [84,85]. Due to the transfer of the ED from the ligands to the metal, after the removal of the *d*_z^2^_ electron, the Mn charge increases by only 0.198 a.u. on going from Cp*_2_Mn^0^ to Cp*_2_Mn^+^ (Table 3). Correspondingly, each Cp* ligand becomes less negative by 0.401 a.u. However, the carbon charges change very little (Table 3). Moreover, the C atoms of both carbocycles and methyl groups become more negative upon ionization of Cp*_2_Mn^0^. The total change in the C(ring) and C(Me) charges in the molecule is −0.044 and −0.037 a.u., respectively. The loss of the ED by the ligands is provided, therefore, entirely by the hydrogen atoms of the methyl groups. Indeed, the total increase of the H positive charges is 0.883 a.u. It is worth mentioning that the larger charge shift (0.044 a.u.) corresponds to the H atoms oriented away from the metal (Table 3). This agrees well with the EDD isosurfaces (Figure 7b,c) which demonstrate, however, that, despite the decisive role of H atoms in the total loss of the ED by the ligands during Cp*_2_Mn^0^ ionization, carbon atoms are also involved in the ED redistribution mechanism. The loss of the π-electron density by the carbocyles, which transfer a negative charge to the metal, is compensated by the ED gain through the C(ring)-C(Me) σ-bonds.

## 3. Materials and Methods

### 3.1. PIE and MATI Experiments

Decamethylmanganocene was purchased from Alfa Aesar and purified by sublimation in the PIE and MATI experiments. All manipulations with the air- and moisture-sensitive compound were carried out under an inert atmosphere in a glovebox. The MATI setup was described earlier [64,86]. The Cp*_2_Mn sample was placed into a stainless steel tube heated up to 170–180 °C. The sublimed vapor-phase organometallics diluted with helium (1.5 bar) was expanded into an evacuated chamber using a heated pulsed valve. The molecular beam selected by a skimmer was directed into a high-vacuum (10^−8^ Torr) ionization chamber where the Cp*_2_Mn neutrals were excited (MATI) or ionized (PIE) by the UV laser radiation. Tunable laser pulses in the 225–235 nm region were formed when doubling the dye-laser frequency (Lambda-Physik, Scanmate UV) by the BBO-I crystal. The Nd:YAG laser (Quanta-Ray PRO-190-10) was used for the pumping of the dye laser. The laser wavelengths were calibrated with a wavemeter (Coherent, WaveMaster). In the PIE experiment, the molecular ions formed by ionization of Cp*_2_Mn were detected by a microchannel plate particle detector of the time-of-flight mass spectrometer. The photoionization mass spectrum revealed an intense molecular ion signal at *m/z* = 325 a.u. (C_20_H_30_Mn^+^). The ion signal at variable laser wavelengths was collected and analyzed. In the MATI experiment, the neutrals were excited by a UV laser pulse to the long-lived ZEKE states. To remove the prompt ions, a pulsed electric field of ~1 V/cm was switched on about 20 ns after the laser pulses. Then, another pulse of the +200 V/cm electric field was applied with a 11 µs delay to ionize the ZEKE-state molecules. The obtained Cp*_2_Mn^+^ ions were detected and analyzed like those in the PIE measurements.

### 3.2. DFT Calculations and Electron Density Analysis

The geometries of the neutral and cationic decamethylmanganocene were optimized at the B3PW91/6-311++G(d,p) level of DFT. The combination of the hybrid B3PW91 functional [87,88] and triple-ζ split valence basis set 6-311++G(d,p) including diffuse and polarization functions on hydrogen and the heavy atoms [89,90] demonstrated earlier good performance when modeling the MATI spectra of bisarene complexes [64,65] and mixed sandwiches [66]. The experimental X-ray structure [68] was used as a starting geometry in the optimization procedure. Harmonic vibrational frequencies were calculated and the results were used to compute the Franck-Condon factors and produce the model MATI spectra. No imaginary frequencies were found. The calculations were carried out with the Gaussian 16 program package [91]. The QT AIM analysis of the ED distribution was performed with the AIM ALL suite [92]. The atomic charges were obtained by integration of ED over the atomic basins using the Promega1 algorithm [93]. The EDD isosurfaces were built as a result of the neutral ED subtraction from the ion ED at the ground-state geometry using the Multiwfn code [94].

## 4. Conclusions

The high-resolution MATI spectroscopy provides unprecedented accuracy in determination of the ionization energy of decamethylmanganocene. The *I* value of 43,146 ± 8 cm^−1^, or 5.349 ± 0.001 eV is obtained on the basis of the Cp*_2_Mn MATI spectrum. The adiabatic and vertical ionization energies of Cp*_2_Mn coincide. The precise *I* value makes it possible to determine with high accuracy the difference between the Mn-Cp* mean bond energies in the Cp*_2_Mn^+^ cation and Cp*_2_Mn^0^ neutral (100.6 ± 0.1 kJ mol^−1^). The bond strengthening in the cation can be associated with a weaker repulsion between the less negative Cp* ligands and stronger Mn-Cp* covalent interactions that resulted from the increase in the d_xz,yz_ electron density. The B3PW91/6-311++G(d,p) model of the MATI vibronic structure agrees very well with the experiment. Although the geometry of both Cp*_2_Mn^0^ and Cp*_2_Mn^+^ is distorted because of the JT effect, only the symmetric Cp*-Mn-Cp* stretching vibration and Mn-C(ring)-C(Me) bending mode are detected in the MATI structure. Correspondingly, the main structural deformations accompanying the electron detachment include an elongation of the Mn-C bonds and changes in the Me out-of-plane bending angle. Interestingly, atomic charge calculations using the QTAIM method predict that the largest part of the reduction in ED during ionization is associated with substituent hydrogen atoms, despite the metallic nature of the ionized orbital. The ED difference isosurfaces reveal, however, a complex mechanism of the charge redistribution involving also the carbon atoms of the molecule. Thus, DFT-assisted MATI spectroscopy represents a powerful instrument that unveils the details of the structural and electronic transformations accompanying ionization of the permethylated metallocene molecule.

## Figures and Tables

**Figure 1 molecules-27-06226-f001:**
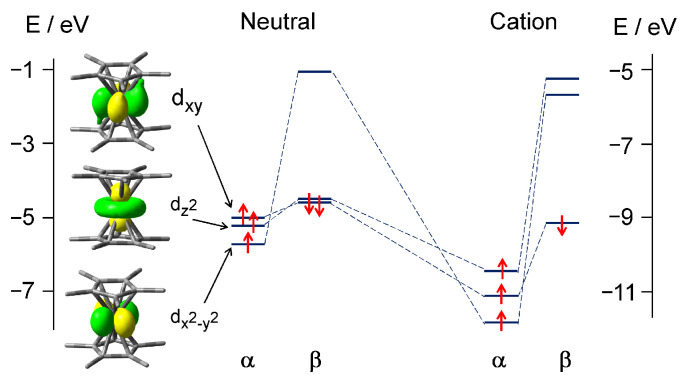
MO correlation diagram of the Cp*_2_Mn^0^ neutral and Cp*_2_Mn^+^ cation. The MO isosurfaces are given for the wavefunction value of 0.04. Notice the different energy scales for Cp*_2_Mn^0^ (left) and Cp*_2_Mn^+^ (right).

**Figure 2 molecules-27-06226-f002:**
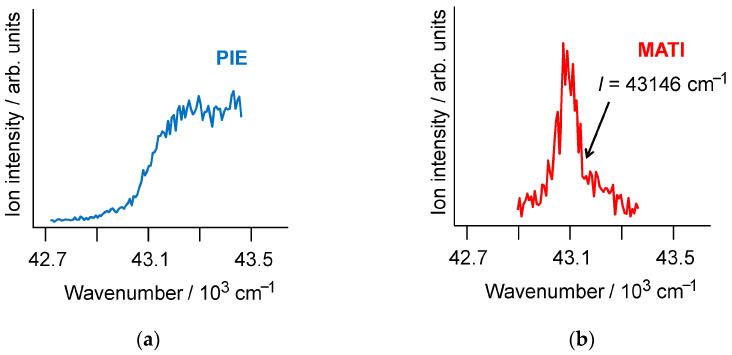
The PIE (**a**) and MATI (**b**) spectra of jet-cooled Cp*_2_Mn. The ionization energy (*I*) corresponds to the narrow range of the MATI signal descend.

**Figure 3 molecules-27-06226-f003:**
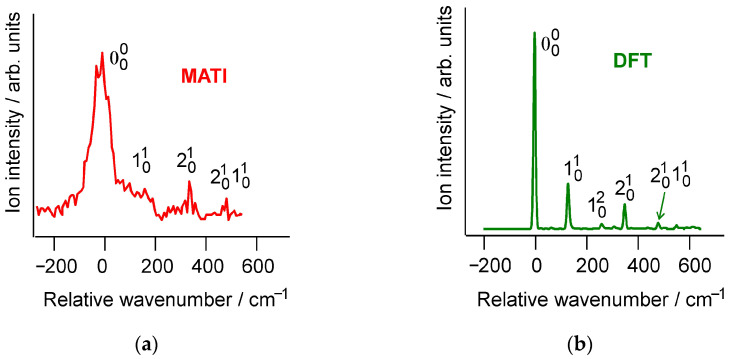
The experimental (**a**) and DFT-simulated (**b**) vibronic structure of the Cp*_2_Mn MATI spectrum.

**Figure 4 molecules-27-06226-f004:**
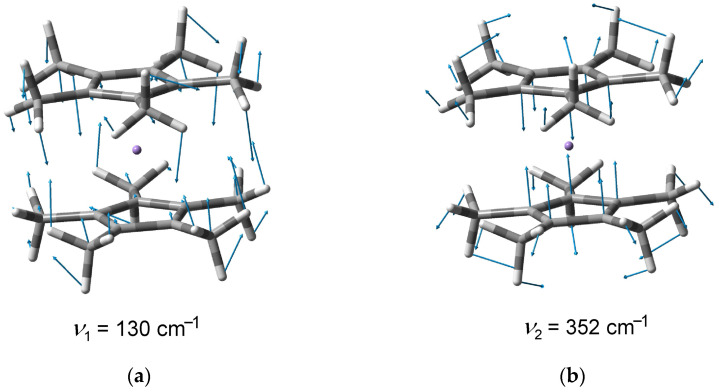
Relative atomic shifts corresponding to the ν_1_ (**a**) and ν_2_ (**b**) vibrations of the Cp*_2_Mn^+^ cation.

**Figure 5 molecules-27-06226-f005:**
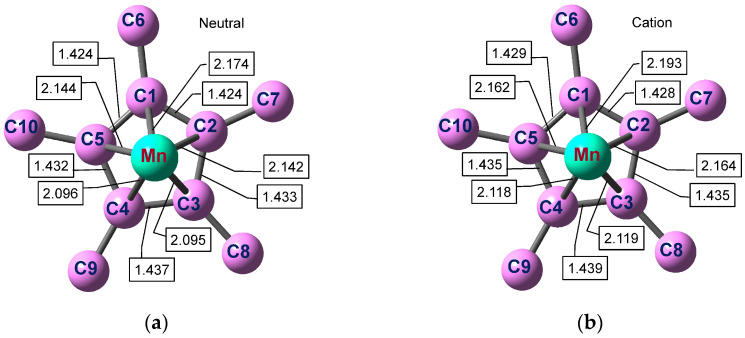
Mn-C(ring) and C(ring)-C(ring) distances (Å) in the optimized Cp*_2_Mn^0^ neutral (**a**) and Cp*_2_Mn^+^ cation (**b**). The hydrogen atoms are omitted for clarity.

**Figure 6 molecules-27-06226-f006:**
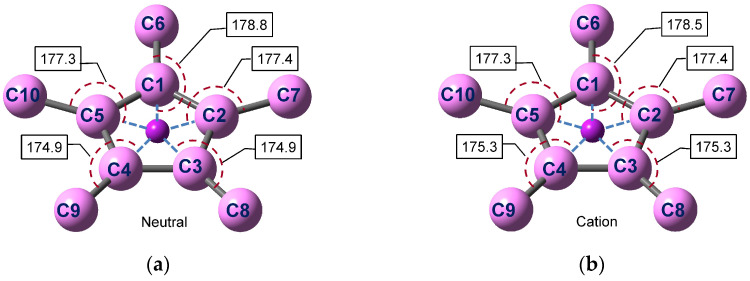
Centroid(ring)-C(ring)-C(Me) angles in the optimized Cp*_2_Mn^0^ neutral (**a**) and Cp*_2_Mn^+^ cation (**b**). The hydrogen atoms are omitted for clarity.

**Figure 7 molecules-27-06226-f007:**
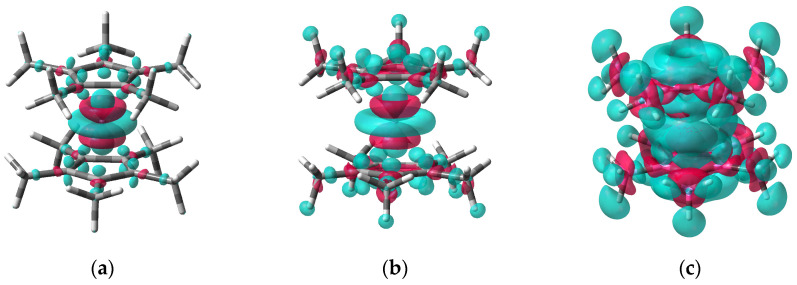
Isosurfaces of the electron density difference between the Cp*_2_Mn^+^ cation and neutral Cp*_2_Mn^0^ at the optimized geometry of the latter (vertical ionization of Cp*_2_Mn). The isovalues are 0.005 a.u. (**a**), 0.003 a.u. (**b**) and 0.001 a.u. (**c**). The blue and red colors correspond to the loss and gain of ED, respectively.

**Table 1 molecules-27-06226-t001:** Relative positions (cm^−1^) of the vibronic components in the Cp*_2_Mn experimental MATI spectrum and its DFT simulation.

Vibronic Component ^a^	MATI	DFT
0_0^0^_	0	0
1_0^1^_	140 ^b^	130
1_0^2^_	-	261
2_0^1^_	340	352
2_0^1^_1_0^1^_	480	482

^a^ The vibronic transitions are notated as N_v″^v′^_, where N is the vibration number and v″ and v′ are the vibrational quantum number in the neutral and ionic state, respectively. ^b^ Shoulder.

**Table 2 molecules-27-06226-t002:** Calculated changes in selected bond lengths and atom-plane(ring) distances ∆*r* (Å) as well as angles ∆ϕ (°) upon ionization of Cp*_2_Mn^0^.

Distance	Δ*r*	Angle	Δϕ
C1-C2	0.004	Mn-centroid(ring)-C1	−0.1
C2-C3	0.002	Mn-centroid(ring)-C2	0.0
C3-C4	0.002	Mn-centroid(ring)-C3	0.1
Mn-C1	0.019	Mn-C1-C6	0.1
Mn-C2	0.022	Mn-C2-C7	−0.3
Mn-C3	0.024	Mn-C3-C8	−0.7
C6-plane(ring)	0.009	centroid(ring)-C1-C6	−0.3
C7-plane(ring)	0.001	centroid(ring)-C2-C7	−0.1
C8-plane(ring)	−0.012	centroid(ring)-C3-C8	0.6
Mn-plane(ring)	0.024		

**Table 3 molecules-27-06226-t003:** Charges *q* (a.u.) on selected atoms ^a^ and fragments in the optimized Cp*_2_Mn^0^ neutral and Cp*_2_Mn^+^ cation and their change upon ionization ∆*q*.

Atom or Fragment	*q* (Neutral)	*q* (Cation)	∆*q*
Mn	0.874	1.072	0.198
C1	−0.103	−0.105	−0.002
C2	−0.114	−0.118	−0.004
C3	−0.123	−0.129	−0.006
C6	0.000	−0.003	−0.003
C7	0.003	−0.001	−0.004
C8	0.005	0.001	−0.004
H(C6) ^b^	0.006	0.050	0.044
H(C6) ^c^	0.005	0.028	0.023
H(C6) ^c^	0.006	0.029	0.023
H(C7) ^b^	0.009	0.053	0.044
H(C7) ^c^	0.007	0.029	0.022
H(C7) ^c^	0.008	0.031	0.023
H(C8) ^b^	0.012	0.056	0.044
H(C8) ^c^	0.009	0.031	0.022
H(C8) ^c^	0.010	0.032	0.022
Me(C6)	0.023	0.108	0.085
Me(C7)	0.027	0.112	0.085
Me(C8)	0.032	0.116	0.084
Cp*	−0.437	−0.036	0.401

^a^ See the text and Figure 5 and Figure 6 for the atom numbering; the carbon atoms of the methyl groups are indicated in parentheses. ^b^ The H atoms oriented away from the metal. ^c^ The H atoms oriented towards the metal.

## Data Availability

The data reported in this study are available in the present article and in the Appendix A.

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
