# Peer review of "Ionization of Decamethylmanganocene: Insights from the DFT-Assisted Laser Spectroscopy"

_molecules, 2022, doi:10.3390/molecules27196226_

Round 1

Reviewer 1 Report

The present manuscript by Ketkov and coworkers is a timely and detailed investigation that provides a very accurate value of the ionization energy of Cp*2Mn and, by virtue of a combination of vibrational excitations accompanying the MATI signal, the structural changes accompanying oxidation, as well as the change in Mn-Cp* binding energy. Structure changes and electron density differences were mapped by DFT calculations which, in spite of the rather demanding intricacies of electronic structure, provided a convincing match with the experiment. The somewhat surprising outcome is that the Cp* methyl substituents carry much of the burden of the ionization in terms of electron density losses. We have experienced a similar situation in some complexes with alkyl-substituted PR3 ligands, so that this may be a more general phenomenon. I gladly recommend the acceptance of this fine manuscript and have only minor corrections or suggestions as detailed below.

Maybe the authors might also say a few words as to what, according to their results, causes the large gain in Mn-Cp* binding energies on ionization (it is not the increase in positive charge at Mn, but rather that of the negative charge at the Cp rings, right?)

Page 2, paragraph 2, line 2: I do not think that the metallocenes have the problem of low volatility, since all of them are easily sublimed

Section 2.2: In this paragraph, the authors should distinguish between experimental and calculated vibrational energies; on first glimpse is seems that the reported values of 130 cm-1 and 352 cm-1 are the experimental ones.

Corrections:

Page 1, Intro: Line 1: on the unique sandwich…; line 4: I suggest to delete the “various” in fundamental and applied science (not sciences): it is not so much the science that varies, but the applications

Page 2, paragraph 2, line 2: remains; line 12: electronic ground state rather than ground electronic state; paragraph 3: the way how the d-orbitals are written, is rather odd; better dx2-y2 etc.

Page 3, paragraph 2, line 2 from bottom: provides an extremely accurate value of the ionization…; paragraph 3, line 1: replace “can give” by “provides”; eqautions (1) and (2): What does the D on the left mean? In eq. 2, replace the “=” by a reaction arrow

Page 8, middle of paragraph 1: The notion that the total change in the charges of the molecule is -0.044 and -0.037 is odd, in particular as Table  indicates a loss of charge from the Me gc atoms. This must be corrected/clarified.

Page 8: The definition of “H atoms on the outer side of the carbocycles” and “located in between the carbocycle planes” are not entirely clear. Better say those who are oriented away from or towards the Mn ion.

Page 4, line 7: replace “takes place” by “holds” or “applies”; line 11: produce MATI specztra (delete”the”; further on: showing the 000 vibronic component as the strongest feature; line 12: vibrationally (two l, typo); section 2.2, line 9: the 352 cm-1 vibration; lines 10/11: replace the “standing unmoved” by maybe “while there is no displacement of the methyl carbon nuclei”; line 12: Both, the n1 and n2 modes, involve

Page 5, line 2 from bottom in paragraph 1: better say “studying the fine details of the structural… ?

Page 6, line 1: beter say: ….of the neutral Cp*2Mn molecule or the corresponding cation…; last line in paragraph 2: singly (not single), lines 1/2 in last paragraph: in a lengthening of

Page 7, section 2.4, end of line 7:  one “the” too many; line 9: Isn´t it rather a shift of electron density from the ring carbon atoms to the methyl carbon atoms?; legend to Fig. 7. and neutral Cp*2Mn

Page 8, line 1, of the ED; line 8:

Page 9, line 2: steel (not still)

Reviewer 2 Report

The manuscript „Ionization of decamethylmanganocene: Insights from the DFT-assisted laser spectroscopy“ by Sergey Ketkov, Sheng-Yuan Tzeng, Elena Rychagova and Wen-Bih Tzeng address one of the most important classes of organometallics with wide prospects for practical use in various fields of chemistry, materials science, molecular electronics, biomedicine etc. and is therefore potentially interesting for a wide range of readers.

The study seems to be performed in a solid way and the presentation seem to me clear and well planned. That I personally dislike pink carbon atoms maybe because I dislike pink in general beside phenolphthalein.  

In citation 89:

Clark, T.; Chandrasekhar, J.; Spitznagel, G.W.; Schleyer, P.V.R.

Has the “v” in P.V.R. to be small one

=> Clark, T.; Chandrasekhar, J.; Spitznagel, G.W.; Schleyer, P.v.R.

Ceterum censeo: publish as it is.

Reviewer 3 Report

The submitted manuscript is nicely written, interesting, clear and concise. However, it also requires some revisions, that I have listed below.

In the introduction, the figure presenting the structural formula of the studied complex should be placed.

In the introduction, more information about the structure of the studied complex should be provided. The Authors should focus on the fact that it undergoes polymorphic phase transition, ( DOI: 10.1002/zaac.19915950105 ).

Line 293, since the crystal structure of the studied complex is well known and has been deposited in CCDC, it should be used as initial for calculations, as it is always better to start from some experimentally proven energetic minimum. Besides, the Authors should use the structures of the two polymorphic forms of the complex as initial to calculations and answer the question whether the choice of initial structure (one of two existing polymorphs) has an influence on the received results. I have a feeling, based on my previous experience with similar complexes, that it may have.

Lines 300-301, why the Authors have not chosen the NBO analysis? I strongly recommend that.

Line 303, it should be “ED”

Lines 298-299, have you found any imaginary frequencies?

Figures 5a and 6a, please compare with crystallographic data from CCDC.

Table 1, why is 102 not observed experimentally?

The Authors have cited a lot (more than 15) of their own works. I think, that this number should be reduced as not all of them are crucial.

Round 2

Reviewer 3 Report

The Authors have made the necessaty corrections.